# Aid when 'there is nothing left to offer': Experiences of palliative care and palliative care needs in humanitarian crises

Lisa Schwartz[1]*, Elysée Nouvet[2], Sonya de Laat[3], Rachel Yantzi[1], Olive Wahoush[4], Wejdan A. Khater[5], Emmanuel Musoni Rwililiza[6,7], Ibraheem Abu-Siam[8], Gautham Krishnaraj[9], Takhliq Amir[10], Kevin Bezanson[11], Corinne Schuster Wallace[12], Oumou Bah Sow[13,14], Alpha Ahmadou Diallo[13,14], Fatoumata Binta Diallo[15], Laurie Elit[16], Carrie Bernard[17,18], Matthew Hunt[19]

1 Health Research, Evidence, and Impact (HEI), McMaster University, Hamilton, Ontario, Canada, 2 School of Health Studies, Western University, London, Ontario, Canada, 3 Global Health, McMaster University, Hamilton, Ontario, Canada, 4 School of Nursing, McMaster University, Hamilton, Ontario, Canada, 5 Adult Health Nursing Department, Faculty of Nursing, Jordan University of Science and Technology, Ar Ramtha, Jordan, 6 University Teaching Hospital of Kigali (CHUK), Kigali, Rwanda, 7 Aarhus University, Aarhus C, Denmark, 8 Migration Studies, Granada University, Granada, Spain, 9 School of Medicine, Queen's University, Kingston, Ontario, Canada, 10 Dalhousie University, Halifax, Nova Scotia, Canada, 11 Northern Ontario School of Medicine, Thunder Bay, Ontario, Canada, 12 Department of Geography and Planning, University of Saskatchewan, Saskatoon, Saskatchewan, 13 Ministère de la santé, Conakry, Guinée, 14 Université de Conakry, Conakry, Guinée, 15 Pédiatrie CHU Donka, Université Gamal Abdel Nasser Conakry, Conakry, Guinée, 16 Obstetrics and Gynecology, McMaster University, Hamilton, Ontario, Canada, 17 Department of Family and Community Medicine, University of Toronto, Toronto, Ontario, Canada, 18 Department of Family Medicine, McMaster University, Hamilton, Ontario, Canada, 19 School of Physical and Occupational Therapy, McGill University, Montreal, Canada

* schwar@mcmaster.ca

**Data Availability Statement:** As this study analyzes qualitative research data, and to respect the consent agreements made with participants,

## Abstract

Access to palliative care, and more specifically the alleviation of avoidable physical and psychosocial suffering is increasingly recognized as a necessary component of humanitarian response. Palliative approaches to care can meet the needs of patients for whom curative treatment may not be the aim, not just at the very end of life but alleviation of suffering more broadly. In the past several years many organizations and sectoral initiatives have taken steps to develop guidance and policies to support integration of palliative care. However, it is still regarded by many as unfeasible or aspirational in crisis contexts; particularly where care for persons with life threatening conditions or injuries is logistically, legally, and ethically challenging. This article presents a synthesis of findings from five qualitative sub-studies within a research program on palliative care provision in humanitarian crises that sought to better understand the ethical and practical dimensions of humanitarian organizations integrating palliative care into emergency responses. Our multi-disciplinary, multi-national team held 98 in-depth semi-structured interviews with people with experiences in natural disasters, refugee camps in Rwanda and Jordan, and in Ebola Treatment Centers in Guinea. Participants included patients, family members, health care workers, and other staff of humanitarian agencies. We identified four themes from descriptions of the struggles and successes of applying palliative care in humanitarian settings: justification and integration of palliative care into humanitarian response, contextualizing palliative care approaches to

the data cannot be shared publicly. Sharing transcripts would violate the agreement to which the participants consented, so we will be able to share only deidentified or anonymized data limited to our consent agreements. The authors make excerpts of the transcripts relevant to the study available within the paper, and in other publications, and summaries on the funders website https://www.elrha.org/?s=Palliative +care&search_post_type=all. Additionally, we have created a page on a public data sharing platform called Borealis. The page related to this website will be accessible here https://doi.org/10.5683/SP3/ 5GOSG3. Through Borealis, this data is available for long-term access on a public platform as required, and additional material may be shared upon request. We have shared data to the extent we can given that the data contain potentially identifying and sensitive patient and participant information, and will continue to add materials as the analyses progress and publications emerge. The Borealis website, our funder's website, and our own program website provide other information which interested researchers may wish to view. These will ensure persistent or long-term data storage and availability, and we confirm that the data will be available to qualified researchers.

**Funding:** This research project is funded by ELRHA's Research for Humanitarian Crisis (R2HC project #19775 to LS, MH) Programme, which aims to improve health outcomes by strengthening the evidence base for public health interventions in humanitarian crises. R2HC is funded by the UK Foreign, Commonwealth and Development Office (FCOD), Wellcome, and the UK National Institute for Health Research (NIHR).

**Competing interests:** The authors have declared that no competing interests exist.

crisis settings, the importance of being attentive to the 'situatedness of dying', and the need for retaining a holistic approach to care. We discuss these findings in relation to the ideals embraced in palliative care and corresponding humanitarian values, concluding that palliative care in humanitarian response is essential for responding to avoidable pain and suffering in humanitarian settings.

## Introduction

Palliative care is an area of growing global concern. During many natural disasters, epidemics, armed conflicts, and mass migrations, humanitarian organizations operate in low resource, high mortality settings. This sphere of action raises crucial questions about how to care for people with life threatening or limiting injuries or conditions. Access to palliation, and more specifically the alleviation of avoidable physical and psychosocial suffering, have been affirmed as fundamental human rights [1–5]. Palliative care is also increasingly recognized as a necessary component of humanitarian practice [6–11].

The latest version of the Sphere Minimum Standards for Humanitarian Response added palliative care as a component of essential healthcare urging that "People have access to palliative and end-of-life care that relieves pain and suffering, maximises the comfort, dignity and quality of life of patients, and provides support for family members" [12]. However, it is still regarded by many as unfeasible or aspirational in crisis contexts; particularly where care for persons with life threatening conditions or injuries is logistically, legally, and ethically challenging. Crises such as the 2014–2016 Ebola outbreak crystallized the need for humanitarian healthcare providers to be better equipped to enact and recognize the value and possibilities of integrating palliative care. The COVID-19 pandemic surfaced challenges and limited preparedness for ensuring dignified end-of-life and adequate symptom control globally, including high- middle- and low-income countries, with conditions familiar in humanitarian crises such as supply chain interruptions, overall strain from the public health emergency on human and non-human health resources, and efforts to limit provider and family risks of infection [13–15].

To better understand the ethical and practical dimensions of humanitarian organizations integrating palliative care into emergency response, we undertook a multicomponent research program with the goal of informing planning, education, and policy. Here we draw on our findings to articulate a vision to strengthen palliative care provision in humanitarian response (we described the obstacles elsewhere [16]).

## Methods

This article presents a synthesis of findings from five qualitative sub-studies within a research program on palliative care provision in humanitarian crises. Drawing on Interpretive Description methodology [17], we used in-depth semi-structured interviews, conducted within a constructivist paradigm [18]. A total of 98 participants were interviewed across the five sub-studies. The research program aimed to develop evidence clarifying ethical and practical possibilities, challenges, and consequences of humanitarian organizations addressing or failing to address patients' and families' palliative needs.

Building on a prior literature review [19], we first interviewed 24 humanitarian health workers and policymakers [16, 20]. These interviews helped us develop four sub-studies focusing on specific contexts of humanitarian action: natural disasters (17 participants), refugee

camps in Rwanda (17 participants), refugee camps in Jordan (21 participants), and in Ebola Treatment Centers in Guinea (16 participants) [21]. We employed purposive and snowball sampling techniques to recruit participants with diverse perspectives on palliative care, including local and international healthcare providers, and non-medical humanitarian agency representatives, as well as patients and family members. Interviews were recorded, translated when necessary, and transcribed.

Within each sub-study, we used inductive data analysis and constant comparative techniques to code and categorize data, and we undertook thematic and targeted analyses [22]. We then created data display tables to compare sub-study commonalities and differences. Five team members independently created lists of key concerns across the sub-studies. These lists were merged and refined through multiple rounds of discussion to develop a set of common themes related to the practical and ethical possibilities of humanitarian organizations providing palliative care during crises.

Ethics approvals were received in each of the locales where interviews were undertaken, in Guinea from le Comité National d'Ethique pour la Recherche en Santé (CNERS) #075/CNERS/17, in Jordan from the Institutional Review Board for the Jordan University of Science and Technology and the King Abdullah University Hospital #37/104/2017, and in Rwanda from the Rwanda National Ethics Committee #97/RNEC/2018. We also received ethics approval from the universities of the Principal Investigators, McMaster University's Hamilton Integrated Research Ethics Board (HiREB) #1855 and McGill University's Institutional Review Board #A10-B54-16B. In addition, we received letters of support from local authorities where participants were interviewed in refugee camps, this includes from the Ministry of Health in Rwanda (#20/............/MIN/2018), and clearance from UNHCR to conduct research in refugee camps in Jordan. All participants provided written informed consent prior to participating in interviews.

## Results

We identified four themes from descriptions of the struggles and successes of applying palliative care in natural disaster, public health emergency, or refugee settings. In the discussion, we consider these findings in light of palliative care ideals and humanitarian values [23, 24]. Participants are quoted verbatim to illustrate aspects of the analysis. Attribution of quotations to participants is by sub-study (Guinea, Jordan, Natural Disasters, Rwanda), whether the participant was a health care worker (HCW), refugee or patient (RP), Humanitarian Agency Worker (HAW), and by the order of participants within the respective study. Thus, Rwanda RP2 is the second patient participant interviewed in the sub-study that took place in Rwandan refugee camps.

## Justification and integration of palliative care

Both the importance and challenges of integrating palliative care in humanitarian response were central findings across each sub-study. Respondents suggested that palliative care could be reframed as a core component of good, compassionate care, fulfilling an ethical obligation to address pain and other forms of suffering in all contexts. Enacting this would involve a significant culture shift and review of humanitarian organizations' policies, training, and operational supports to enable teams on the ground to provide quality care inclusive of palliation. As one healthcare provider in a refugee camp explained:

> "I think it's an obligation because, . . . even if we know that they will not get cured, they need assistance, and they are human beings, they are still alive, . . . [health professionals]

have to take care of them, they have to provide them what they need until the last minute, they cannot reject them from getting care." (Rwanda HCW04)

There were some cautions raised, including the importance of clearly defining palliative care as only one part of the wider continuum of care, acknowledging the extent to which it can be fully realized is context dependent, and asserting that ethically it must never be substituted for providing the best possible curative care. As one participant cautioned, aid workers must not lose sight of their role as advocates, "because what you should be there doing is trying to prove to anyone who will watch or listen that these are lives that can be saved." (Natural Disasters HCW3).

Others expressed that palliative care might not be happening because people do not understand it or dismiss it as unfeasible and inappropriate. Such concerns were also reflected by some participants who were refugees. They expressed that when basic needs are not met everything else is secondary, including palliative care. As a refugee living in a community setting in Jordan (not in a refugee camp) put it: "Only basic life needs, that's all what I need." (Jordan RP1).

It was stressed that to provide palliative care requires advocating for access to pain medications including opioids, accounting for concerns about correct use, appropriate regulation and training, and attending to cultural considerations. However, pain and symptom management needed to be combined with interdisciplinary care:

"The other side is the medication, all the pain relief, things like that. But I say, the mental health counselling, the psychiatric side counts for a lot. A lot." (Rwanda HCW05)

Ultimately, the commitment to offering palliative interventions was centred on notions of equitable access to treatment, and non-abandonment of patients. An international physician responding to an earthquake emphasized impartiality as a justification for providing palliative care during a crisis:

"... the palliative aspect of even for the immediately dying person in the disaster is of utmost importance. To me it's as valuable–if you want to treat people equally–it's as valuable as treating someone who's going to survive." (Natural Disasters HCW07)

Respondents emphasized that planning for palliative care should be mainstreamed in humanitarian response, as ad hoc efforts have proven inconsistent, uncoordinated, or nonexistent. Such planning was described as requiring training, clear protocols, and ensuring access to necessary supplies and medications. Participants stressed that doing so demonstrates the humanitarian commitment to respond to suffering, dignity, and non-abandonment throughout care, including at the end of life.

## Contextualized to crisis

More than one approach is needed to integrate palliative care in different types of humanitarian contexts. Attention must be paid to the type of crisis, the capacities of international and national organizations involved, and the realities of local health care systems.

While the key components of palliative care remain constant, the type of crisis determines available lead time, duration, numbers of people affected, types of injuries and illnesses. It also affects the extent to which existing healthcare systems and supply chains are damaged or overwhelmed.

Broadly speaking, practical restrictions arising from humanitarian crises limit organizational capacities. These limitations span availability of transportation, bureaucratic structures, integration with local health systems, and design of physical spaces. Respondents indicated some of these could be improved through palliative perspectives to care, such as in the design of a refugee camp:

"Even with wheelchairs they can't enter where they are living, the doors are too small. . . . That's why I was saying palliative care is wide, you have to look at the shelters, the infrastructure, everything, the landscape of the camp" (Rwanda HCW07)

Limited integration between humanitarian interventions and national healthcare systems in longer term refugee settings resulted in patients having to navigate fragmented services and bureaucracy. Facing labyrinthine care pathways and complications of being cut-off from familiar resources, patient-respondents wished for reduced bureaucratic delays and better continuity of care. As one refugee-respondent in Jordan explained ". . . it's a long, complicated procedure and routine which needs a lot of paperwork" (Jordan RP5), while another expressed that "what I am afraid of is that my situation will become worse due to long delays" (Jordan RP6). This fear was realized for a third patient-participant whose condition worsened as paperwork took weeks to be resubmitted.

Tensions also emerge where the humanitarian system interfaces with national health systems. In regions where palliative care was only available through humanitarian agencies, this could create problems. For example, in some refugee settings,

"we don't prefer to provide services to the refugees at a level higher than what the normal or the local government is providing the nationals because this creates a social tension" (Jordan HAW01)

In Ebola Treatment Centers in Guinea, health professional participants described patient care being shaped by infection control measures and isolation requirements. These required the use of personal protective equipment and limited the time spent with patients. The high mortality rate from Ebola and architecture of Ebola Treatment Centers also resulted in exposure of the critically ill in some centers to distressing views of the dead:

"Every day they wrapped the dead. All they did was wrap the dead, only wrapping, only wrapping, only wrapping. He! The. . . door would not close. The rooms were stuck together, and we saw everything through the door." (Guinea RP5).

Natural disasters and conflict settings brought other realities. Triage in emergency and mass casualty settings creates its own exigencies, and a healthcare provider who responded to the 2010 earthquake in Haiti reported,

"it forces you to suppress that human side of you. . .But the emergency part is you have to do all the best you can do to save the maximum people you can save. So, you have to be a little bit not human. When I say 'not human' is to see someone that you know is gasping you know that person is going to die so let me take the other one." (Natural Disasters HCW11)

Many reported that palliative care is easily forgotten during an acute emergency. A healthcare provider in Bangladesh remarked that at the acute phase of the crisis they thought there were no patients with palliative care needs. Once they started looking, "they were everywhere"

(Natural Disasters HCW21). Respondents indicated that families may not be present or available in disasters or may have predeceased the patients in need of palliative care. Such patients were identified as *the vulnerable of the vulnerable* at risk of being neglected or forgotten in a crisis, so respondents emphasized an ethical obligation to address pain and other forms of suffering even in emergency contexts.

To tailor the integration of palliative care to crisis settings and local health system contexts, planning should address interactions and compatibility between aid-based resources and national resources. This could work well with responses that involve the possibility of harmonizing approaches and modifying interventions to ensure timely and practical access for all.

### Attentive to the "situatedness of dying": local social-cultural and structural considerations

Participants described the need for attention to features such as religious values, mourning rituals, beliefs about pain and opioids, as well as local languages.

> "If you're gonna do palliative care the right way it needs to be done with a lot of cultural knowledge and sensitivity.. . . Impose your views and that's sometimes challenging, I've seen that a lot in missions. . . . So, we have to be very knowledgeable. I think that's one of the first things we should do when we get there on a mission. Know about, among other things, what are their rites, what do they do when somebody dies. What's culturally appropriate and then it's up to us to follow that." (Natural Disasters HCW02)

Several participants emphasized that local care providers are best positioned to clarify patient and community expectations and beliefs. However, care is often delivered at least in part by international providers and agencies, requiring effective communication and integration of care. Humanitarian crises often challenge local means of supporting death and dying (e.g., funeral practices during infectious disease outbreak), while displacement and social fragmentation can further impede these processes:

> ". . . they're worried about the foreign country. . . worried about dying in the camp, but outside the camp, they say maybe I will not even get a coffin, they're worried about many things. And also, the family, how my children, my husband, my sister, can get from [country of origin] up to here to bury me . . .It's very painful." (Rwanda HCW04)

In Guinea, a major emphasis from participants was the possibility of helping people "die in honour" which consisted of being accompanied, being able to share final messages with loved ones, and enacting cultural practices related to the bodies of the deceased.

Local strategies emerged to help mitigate disruptions to norms around end-of-life care during a crisis. For example, Ebola survivors were recruited to provide psychosocial care in Guinean Ebola Treatment Centers, even simply to hold a hand. In Rwandan refugee camps, a local network of fellow refugees called the Good Samaritans helped with daily life and provided care to suffering individuals. While such measures cannot replace the presence of dislocated, absent, or lost loved ones and homes, they were described by participants as invaluable sources of support. Nevertheless, these informal local strategies do not exist everywhere and have not been formalised. HCW respondents asserted that with good training and a proper budget, these volunteers could greatly strengthen palliative care.

Differences in expectations and practices around disclosure of a terminal diagnosis posed particular challenges in several contexts. For example, international health professionals'

expectations to share prognoses and discuss plans with patients and families did not always align with local cultural norms. Some HCW participants argued it was best that patients are not informed of their prognosis, or of the loss of their loved ones while the patient was acutely ill or injured. The sense from those participants was that disclosure could provoke despair, reducing patients' will to fight, especially where outcomes were uncertain. On the other hand, as one international participant queried, how can you "... establish firmly and convincingly that your patient is choosing a palliative pathway without a direct conversation, I don't know how to do that." (Natural Disasters HCW03) Even if prognosis and palliative status were not fully discussed with patients due to cultural and family preferences, it did not prevent care providers implementing a palliative approach.

Collaboration and consultation are necessary to facilitate safe and contextually appropriate palliative care, discussions of prognosis with patients and families (where present), bereavement and funeral practices wherever possible, and engagement with local leaders to identify new practices (e.g. adapted funeral rites during Ebola). Participants underlined the critical role of local lay and professional healthcare providers in guiding teams affiliated with international organizations and collaborating to provide culturally congruent care.

## Holistic

Particularly given the impacts of crises on community and family, the narratives of participants underscore the importance of being present with people in their suffering. In this sense, palliative care was understood as a concrete action to not abandon patients, especially whose lives could not be saved. Social isolation, including being separated from families, dying in a foreign country, cut off from ancestral connections, and/or having no advocate or support, generally seemed to contribute to worse outcomes. This led to an articulated need for humanitarian organizations to help maintain patient dignity through providing assurance to families about the care of their loved one in dying and death.

Fear of dying unaccompanied increased the suffering of individuals in Ebola Treatment Centres and refugee settings. Fear was compounded for those who felt exposed to stigma by association with infectious diseases like Ebola, or non-communicable diseases like cancer. A participant in a refugee setting described how his belongings were taken by others while he was at the hospital, as if they considered him to already be dead. Having family or an advocate for support helped give patients a voice, though, for some, this was not possible: "...to not be with one's family, to realize the burial will be different: this was hard, and maybe even harder for the surviving family" (Rwanda HCW16). A comprehensive approach to palliative care needs to also address these social dimensions of suffering.

Participants described psychosocial care, for patients and for their families, as an important feature of a holistic approach to palliative care. Families' needs extend to support of grieving before, during and after someone has died. For example, parents wanting to ensure their children will be cared for when they are gone, prompting collections for school uniforms and milk money to reassure two dying refugee mothers. In another narrative, after a stillbirth,

> "...the mother she wouldn't look at her baby at first. So I held him and I said is it ok if I touch him. She said yeah but she was crying and telling me it's not fair and all that. So, I let her express all that. And I started touching him ... and I started saying how beautiful he was. She sort of looked at me curiously and she put her hand on mine. She wouldn't touch him directly, ... and she put her hand on mine at first and she ended up holding him." (Natural Disasters HCW2)

Across all the study settings, actions that address "small things" were seen as extremely valuable by both patients and care providers. Some patient participants expressed a simple desire to connect with family in order to say their goodbyes, while a refugee in Jordan stated their primary concern was "If someone could just bring the water to my tent." (Jordan RP02). Providing shade from the hot sun, nutritious food, making childcare available, or building a coffin, were all identified as important means of alleviating suffering and demonstrating compassion. A participant in Rwanda expressed,

> "I am told that my illness will not heal. As it is hard to do household works, I would wish to have some help. As long as I will still be alive and as my kids will need me, I would need someone to help me prepare food for them and wash their clothes. . ." (Rwanda RP01)

Holistic care requires intentional training and inclusion of both clinical and psychosocial aspects of palliative care to be realized, and by extension reinforces the importance to support local responses wherever possible.

## Discussion

Integrating palliative care in humanitarian contexts engenders fundamental questions about the role of humanitarian action and the aims of humanitarian healthcare. The four themes considered alongside principles of palliative care and principles of humanitarian action demonstrate values in tension. Palliative care is patient and family centred, integrative, comprehensive, and ideally involves continuity [25] Humanitarian values are responsive, rescue driven, and public health oriented [23]. Humanitarian response is a broad system of interactions; engaging with the micro systems of individuals and families is only one part of the activities of organizations. Nevertheless, palliative care and humanitarian action also have common ground in commitment to the human right of dignity and humanitarian assistance, response to suffering, and non-abandonment [12].

Based on the interviews, and corroborating a growing emphasis in the literature, it is apparent that humanitarian healthcare responses, including triage, need to include at minimum accompaniment and comfort focused care to the extent possible. Otherwise, a focus reserved to 'life saving' actions facilitates what Smith and Aloudat identified as a 'false dichotomy' [11], wherein humanitarian actors engage in *either* curative care *or* palliative care and not both, undermining the humanitarian imperative to save lives *and* alleviate suffering. Our findings suggest a critical re-evaluation for humanitarian actors of what counts as success and failure in relation to death and dying, and to examine where and why palliative care can be even more present in the care they provide.

Palliative care need not involve grand interventions; small gestures and acts of kindness were noted in the interviews as having an outsized and tangible impact. Findings show that small things can be a relatively inexpensive way to integrate palliative care and has potential when futile treatment is not pursued to reduce burdens to patients. Humanitarian actors "can provide [critical] drops of humanity" [24] (p1542) as palliative interventions [7], and promote the dignity of all patients, which aligns explicitly with stated humanitarian values.

While these minimum elements are a beginning, our research acknowledges inclusion of palliative care will require more intentional and sustained efforts. Such efforts are justified from an equity informed perspective [26]. Encountering patients with serious health related suffering or at end of life is ubiquitous in humanitarian response. Many participants advocated for improved equitable access to services and basic levels of healthcare for such patients in crisis events. Palliative care specifically targets inclusion of patients and families who are

especially vulnerable amongst those already experiencing enhanced need [27]. In doing so it can be part of efforts to ensure adequate care, and that basic human material, social, psychological, and spiritual needs are being met more widely.

Palliative care in humanitarian contexts is complicated by the reality of global inequities, that while palliative care may be the only option, in other places with better resources many similar patients would be offered curative treatments. Given this reality, there is always a need for concurrent advocacy for effective life-saving treatment along with palliative options and health system strengthening as part of sustainable development and disaster risk reduction [27]. The nature of humanitarian response and inherent resource constraints mean that end of life needs are unavoidable. Thus, palliative care must be both intentionally integrated into healthcare responses during crises and adequately resourced. Its impact will be marginalized if it is treated as a strictly separate focus or program, with reasoning that it is "nice but not necessary," [28]. Rather palliative care—incorporating coordinated multisectoral responses (shelter, food, wash, health etc.) and mainstreaming of health interventions not only at secondary level of care but also at community health and primary health care levels—will ensure those who need it can access essential resources. Integrating palliative care will require additional but modest investment to ensure adequate policy, team structures, education and training, medications, and supplies. Ultimately additional specialized and focused resources will be needed [8].

The depth and complexity of responses from our respondents also highlight the critical need for humanitarian organizations to adapt and frame the principles and components of palliative care in relation to local social, cultural, and spiritual contexts. Death and dying are culturally specific and must be responded to as such [8]. While not unique to palliative care, such adaptation and framing is especially salient and requires collaboration with local care providers to be done effectively.

Finally, reframing non-curative treatment as an enactment of humanitarian principles related to alleviating suffering and promoting dignity can reassure distressed responders [29]. Improving comfort by improvising ways to elevate patients into a reclined position using crates, holding hands, and other simple acts of care and compassion make a real difference, and help providers feel more empowered to provide good care. In addition, organizations should take steps to provide staff support, including training, counselling, and debriefing to help manage moral distress and burn-out [30].

## Limitations and ways forward

While our study design incorporated five in-depth sub-studies, enabling us to look closely at palliative care in multiple humanitarian contexts, we did not include an acute conflict setting amongst these sub-studies. Such contexts present distinct considerations for ethics and care delivery [31], and thus the lack of such a sub-study represents a key limitation. An additional limitation relates to the multi-lingual nature of the project. Interviews were conducted in multiple languages and translated into English for analysis. Translations were reviewed for accuracy but present a limitation regarding fidelity of meaning.

Pathways forward for palliative care in complex humanitarian settings should include but not stop at time and resource investments. Apart from investment, further research would be helpful, including surveys, resource evaluation, and post-implementation studies that would help quantify the impact of palliative care and evaluate the benefits to patients, communities and care providers.

## Conclusion

Our exploration affirms that intentionally addressing the alleviation of suffering for patients and families with life-threatening illness or injury is fundamentally consistent with humanitarian values and practice. Inclusion of palliative care in humanitarian response enables more holistic and integrated care, preserving commitments to equity, dignity, alleviation of suffering and ideals of non-abandonment. A continued shift is therefore needed to reframe palliative care from 'care when we have nothing left to offer', to being understood as enacting core commitments to humanitarian principles. The complexities of implementation are not to be underestimated. However, until palliative care is intentionally and adequately resourced as an essential part of humanitarian healthcare, preventable suffering for patients, families, and providers will continue.

Humanitarian organizations are taking new steps toward integrating the palliative principles and approaches outlined in Sphere and WHO guidelines and re-examining practical, operational components to ensure inclusion [32]. As they have always done with other aspects of care, they will have to continue to grapple with how best to implement palliative care in the constrained realities of the most challenging care environments. But the progression from acknowledgments to action is critical.

## Acknowledgments

We would like to express our sincere appreciation to those who participated in our study, particularly those participants who were ill or whose family members were ill. Special thanks too to Marie-Charlotte Böuesseau, Joan Marston, Christian Ntzimara, Erik Krakauer, Doris Schopper, Paul Bouvier for their support and guidance on this work. We also want to thank our collaborators and supporters in Guinea, Jordan, Rwanda, Bangladesh, and elsewhere. In writing this article, we are indebted to and inspired by the memory of Sékou Kouyaté, talented anthropologist and Humanitarian Health Ethics group member, deceased December 16, 2020. This research project is funded by ELRHA's Research for Humanitarian Crisis (R2HC project #19775 to LS & MH) Programme, which aims to improve health outcomes by strengthening the evidence base for public health interventions in humanitarian crises. R2HC is funded by the UK Foreign, Commonwealth and Development Office (FCOD), Wellcome, and the UK National Institute for Health Research (NIHR). Visit elrha.org for more information about Elrha's work to improve humanitarian outcomes through research, innovation, and partnership.

## Author Contributions

**Conceptualization:** Lisa Schwartz, Elysée Nouvet, Sonya de Laat, Olive Wahoush, Wejdan A. Khater, Emmanuel Musoni Rwililiza, Ibraheem Abu-Siam, Gautham Krishnaraj, Kevin Bezanson, Corinne Schuster Wallace, Oumou Bah Sow, Alpha Ahmadou Diallo, Laurie Elit, Carrie Bernard, Matthew Hunt.

**Data curation:** Lisa Schwartz, Elysée Nouvet, Sonya de Laat, Rachel Yantzi, Wejdan A. Khater, Corinne Schuster Wallace, Matthew Hunt.

**Formal analysis:** Lisa Schwartz, Elysée Nouvet, Sonya de Laat, Rachel Yantzi, Olive Wahoush, Wejdan A. Khater, Emmanuel Musoni Rwililiza, Ibraheem Abu-Siam, Gautham Krishnaraj, Takhliq Amir, Kevin Bezanson, Corinne Schuster Wallace, Oumou Bah Sow, Alpha Ahmadou Diallo, Fatoumata Binta Diallo, Laurie Elit, Carrie Bernard, Matthew Hunt.

**Funding acquisition:** Lisa Schwartz, Elysée Nouvet, Sonya de Laat, Gautham Krishnaraj, Kevin Bezanson, Oumou Bah Sow, Matthew Hunt.

**Investigation:** Lisa Schwartz, Elysée Nouvet, Sonya de Laat, Rachel Yantzi, Olive Wahoush, Wejdan A. Khater, Emmanuel Musoni Rwililiza, Ibraheem Abu-Siam, Gautham Krishnaraj, Kevin Bezanson, Corinne Schuster Wallace, Oumou Bah Sow, Fatoumata Binta Diallo, Matthew Hunt.

**Methodology:** Lisa Schwartz, Elysée Nouvet, Sonya de Laat, Olive Wahoush, Emmanuel Musoni Rwililiza, Gautham Krishnaraj, Kevin Bezanson, Corinne Schuster Wallace, Laurie Elit, Matthew Hunt.

**Project administration:** Lisa Schwartz, Elysée Nouvet, Sonya de Laat, Rachel Yantzi, Corinne Schuster Wallace, Matthew Hunt.

**Resources:** Lisa Schwartz.

**Software:** Corinne Schuster Wallace.

**Supervision:** Lisa Schwartz, Olive Wahoush, Wejdan A. Khater, Kevin Bezanson, Oumou Bah Sow, Matthew Hunt.

**Validation:** Rachel Yantzi, Emmanuel Musoni Rwililiza, Ibraheem Abu-Siam, Kevin Bezanson, Oumou Bah Sow, Alpha Ahmadou Diallo, Fatoumata Binta Diallo, Laurie Elit, Carrie Bernard.

**Visualization:** Sonya de Laat.

**Writing – original draft:** Lisa Schwartz, Kevin Bezanson, Matthew Hunt.

**Writing – review & editing:** Lisa Schwartz, Elysée Nouvet, Sonya de Laat, Rachel Yantzi, Olive Wahoush, Wejdan A. Khater, Emmanuel Musoni Rwililiza, Ibraheem Abu-Siam, Gautham Krishnaraj, Takhliq Amir, Kevin Bezanson, Corinne Schuster Wallace, Oumou Bah Sow, Alpha Ahmadou Diallo, Fatoumata Binta Diallo, Laurie Elit, Carrie Bernard, Matthew Hunt.

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
