## [Decision Letter · Decision Letter 0]

7 Oct 2022

PGPH-D-22-01377

Aid when ‘there is nothing left to offer’: Experiences of palliative care and palliative care needs in humanitarian crises

Dear Dr. Schwartz,

Thank you for submitting your manuscript to PLOS Global Public Health. After careful consideration, we feel that it has merit but does not fully meet PLOS Global Public Health’s publication criteria as it currently stands. Therefore, we invite you to submit a revised version of the manuscript that addresses the points raised during the review process.

We look forward to receiving your revised manuscript.

Kind regards,

Stephen J. McCall, DPhil

Academic Editor

Journal Requirements:

1. Please indicate the full and correct funding information for your study and confirm the order in which funding contributions should appear.

2.We ask that a manuscript source file is provided at Revision. Please upload your manuscript file as a .doc, .docx, .rtf or .tex.

3. In the online submission form, you indicated that [Insert text from online submission form here]. All PLOS journals now require all data underlying the findings described in their manuscript to be freely available to other researchers, either 1. In a public repository, 2. Within the manuscript itself, or 3. Uploaded as supplementary information.

Additional Editor Comments (if provided):

Reviewers' comments:

Reviewer's Responses to Questions

**Comments to the Author**

1. Does this manuscript meet PLOS Global Public Health’s publication criteria? Is the manuscript technically sound, and do the data support the conclusions? The manuscript must describe methodologically and ethically rigorous research with conclusions that are appropriately drawn based on the data presented.

Reviewer #1: Yes

Reviewer #2: Yes

2. Has the statistical analysis been performed appropriately and rigorously?

Reviewer #1: N/A

Reviewer #2: N/A

3. Have the authors made all data underlying the findings in their manuscript fully available (please refer to the Data Availability Statement at the start of the manuscript PDF file)?

Reviewer #1: Yes

Reviewer #2: Yes

4. Is the manuscript presented in an intelligible fashion and written in standard English?

Reviewer #1: Yes

Reviewer #2: Yes

5. Review Comments to the Author

Reviewer #1: Dear authors,

Thank you for submitting your article to the journal. You describe the results of five qualitative sub-studies on the provision of palliative care in humanitarian crises. You have explored different dimensions by conducting, in different countries, 98 semi-structured interviews with people who have lived experiences of natural disasters, in refugee camps and/or of Ebola outbreaks. Participants included patients, family members, healthcare professionals workers and other staff members of humanitarian agencies. You have identified different themes and conclude that palliative care in humanitarian response is essential to address preventable pain and suffering in humanitarian assistance settings.

I would like to congratulate you for this excellent, relevant and interesting work. The text is well written, easy to follow, even if it is long, especially the results section. Some parts can be moved to an appendix, which I don't think is essential.

I have two other minor comments for the discussion.

1. I wonder to what extent the authors think that there is an imperative need for "a critical re-evaluation for humanitarian actors of what counts as success and failure in relation to death and dying, and to examine where and why palliative care is absent from the care they provide."

The reader may think that it's totally absent from the approach, both at HCW and system levels. If not true, you may wish to rephrase.

2. Apart investment, further research may be helfpul. It looks like quantitative methods may be used, surveys, resource evaluation and post-implementation studies, among other. I suggest adding a perspective paragraph.

Reviewer #2: This is a hugely important paper bringing together a set of evidence on the need for palliative care and the challenges of palliative care delivery in humanitarian settings. By examining the tensions between the principles and practice of palliative care and the principles determining humanitarian responses the paper has provided lever points where actions need to be directed in order to reconcile tensions and create new pathways of work The paper is well written with a clear methodology, appropriately referenced, comprehensive and intentional in understanding of the need to integrate palliative care into systems

6. PLOS authors have the option to publish the peer review history of their article (what does this mean?). If published, this will include your full peer review and any attached files.

**Do you want your identity to be public for this peer review?** For information about this choice, including consent withdrawal, please see our Privacy Policy.

Reviewer #1: **Yes: **Patrice Forget

Reviewer #2: **Yes: **Liz Grant

---

## [Editor Report · Decision Letter 1]

11 Nov 2022

Aid when ‘there is nothing left to offer’: Experiences of palliative care and palliative care needs in humanitarian crises

PGPH-D-22-01377R1

Dear Dr. Schwartz,

We are pleased to inform you that your manuscript 'Aid when ‘there is nothing left to offer’: Experiences of palliative care and palliative care needs in humanitarian crises' has been provisionally accepted for publication in PLOS Global Public Health.

Best regards,

Stephen J. McCall, DPhil

Academic Editor